# Analysis of Clinical Characteristics of Patients with Recurrent Cytomegalovirus Retinitis after Hematopoietic Stem Cell Transplantation

**DOI:** 10.3390/jpm13040639

**Published:** 2023-04-07

**Authors:** Xiaona Wang, Yao Lu, Haiping Li, Zhizhong Ma, Jing Hong, Changguan Wang

**Affiliations:** Department of Ophthalmology, Beijing Key Laboratory of Restoration of Damaged Ocular Nerve, Peking University Third Hospital, Beijing 100191, China; 15801635367@163.com (X.W.); hongjing196401@163.com (J.H.)

**Keywords:** hematopoietic stem cell transplantation, cytomegalovirus retinitis, recurrence, wide-angle fundus photography, optical coherence tomography, polymerase chain reaction, CD4

## Abstract

Objective: To analyze and summarize the clinical and imaging characteristics of patients with cytomegalovirus retinitis (CMVR) relapse after hematopoietic stem cell transplantation (HSCT). Methods: This retrospective case series study recruited patients with CMVR after HSCT. The study compared the patients with stable lesions and CMV-negative aqueous humor after treatment with those with relapse lesions and a CMV DNA load in aqueous humor which had increased again after treatment. The observation indexes were basic clinical information, best-corrected visual acuity, wide-angle fundus photography, optical coherence tomography (OCT), blood CD4^+^ T lymphocyte count, and aqueous humor CMV load of the patients. We summarized the data and statistically analyzed the differences between the relapse and non-relapse groups, as well as the correlations of the observed indicators. Results: The study recruited 52 patients with CMVR (82 eyes) after HSCT, of whom 11 patients (15 eyes) had recurrence after treatment (21.2%). The recurrence interval was 6.4 ± 4.9 months. The final best-corrected visual acuity of recurrent patients was 0.3 ± 0.3. The number of CD4^+^ T lymphocytes in recurrence patients at the time of onset was 126.7 ± 80.2/mm^3^. The median CMV DNA load detected in aqueous humor at the time of recurrence was 8.63 × 10^3^ copies/mL. There was a significant difference in the CD4^+^ T lymphocyte count between the recurrence and the non-recurrence groups at onset. The onset of visual acuity in recurrence patients was significantly correlated with final visual acuity and recurrence lesion area. The fundus of recurred CMVR showed increased marginal activity of the original stable lesion. Concurrently, yellow-white new lesions appeared around the stable, atrophic, and necrotic lesions. OCT showed new diffuse hyperreflexic lesions in the retinal neuroepithelial layer near the old lesions. Inflammatory punctate hyperreflexes were observed in the vitreous, with vitreous liquefaction and contraction. Conclusion: This study suggests that the clinical features, fundus manifestations, and imaging features of CMVR recurrence after HSCT are different from those at the initial onset. Patients should be closely followed up after their condition is stable to be alert for CMVR recurrence.

## 1. Introduction

Cytomegalovirus (CMV) infection remains an important cause of morbidity and mortality in patients who undergo hematopoietic stem cell transplantation HSCT, because the estimated seroprevalence for blood and organ donors is 90% in China [1,2,3,4]. Cytomegalovirus retinitis (CMVR) is the most common fundus disease in people with low immune function [5,6,7,8]. Typical fundus manifestations of the disease are yellow-white fusion lesions that progress along the retinal vessels, accompanied by retinal hemorrhage and edema, showing a typical “cheese and tomato sauce” appearance. Systemic and intravitreal injections of ganciclovir have become a recognized method of treatment of CMVR [9].

However, in clinical practice, we found that patients with different immune reconstitution statuses after HSCT had different disease outcomes. Not all CMVR lesions are stable and can be maintained after aqueous humor polymerase chain reaction (PCR) detection turns negative. In recent years, domestic and foreign literature has gradually paid increased attention to the diagnosis and treatment of CMVR after HSCT; however, there are no reports on the clinical characteristics and imaging manifestations of patients with recurrence after CMVR treatment and HSCT. Therefore, this study analyzed and summarized the clinical characteristics and imaging changes in these patients, aiming to provide a basis for optimization of the diagnosis and treatment of CMVR after HSCT.

## 2. Materials and Methods

### 2.1. Data Collection

This retrospective case series study included patients who were diagnosed with CMVR after HSCT and recurrence after treatment at the Ophthalmology Center of Peking University Third Hospital from January 2018 to October 2022. Inclusion criteria: (1) History of HSCT; (2) CMVR diagnosis according to typical fundus manifestations and positive CMV DNA in aqueous humor with deoxyribonucleic acid (DNA) PCR testing; (3) Patients whose previous CMVR lesions were stable and aqueous humor CMV DNA was negative, who showed a recurrent CMVR lesion and recurrence of aqueous humor CMV DNA positivity. Exclusion criteria: Patients with a history of other systemic and eye diseases, who had undergone other eye surgery or laser treatment in the past, and who could not cooperate with regular follow-up and treatment, were excluded from the study. This study was approved by the Ethics Committee of Peking University Third Hospital. Ethics approval number: 2015; Medical Ethics Review number: 197. The study followed the principles of the Declaration of Helsinki.

The medical history and basic information of the enrolled patients were collected, including gender, age, primary disease, affected eyes, date of hematopoietic stem cell transplantation, date of ocular onset, date of ocular recurrence, peripheral blood CD4^+^ T lymphocyte count, and the results of ocular aqueous humor virus examination at the time of recurrence. The CMV DNA in aqueous humor was detected by PCR, and the CD4^+^ T lymphocyte count in the patients was detected by flow cytometry.

Follow-up observation was carried out on the CMVR lesions of the patients. CMVR lesion locations were divided into 3 zones: zone 1 consisted of the area within 3000 mm of the center of the macula (macular involvement) or within 1500 mm of the margin of the optic disc (disc involvement), zone 2 consisted of the area anterior to zone 1 to the equator, and zone 3 consisted of the area anterior to zone 2 to the ora serrata [5]. Image J software was used to measure the lesion and optic disc areas of the patients with recurrence, and the ratio of the lesion area to optic disc area was recorded.

### 2.2. Follow-Up and Treatment Plan

After the initial diagnosis of CMVR, all patients underwent regular intravitreal antiviral therapy until the lesion was completely stable and CMV DNA tests of the aqueous humor were negative. Thereafter, the patients were followed up once a month for the required eye examinations, including best-corrected visual acuity (BCVA), intraocular pressure, slit lamp examination, ophthalmoscopy after mydriasis, wide-angle fundus photography (Optos PLC, Dunfermline, UK), and optical coherence tomography (Heidelberg Engineering, Heidelberg, Germany). According to the fundus examination, if the lesions were suspected to have recurred, PCR testing of the aqueous humor was performed to confirm CMVR recurrence, and the antiviral treatment was reinitiated.

All patients were administered an intravitreal injection of ganciclovir immediately after the CMVR diagnosis was confirmed. The dosage of ganciclovir was 3 mg/0.1 mL by intravitreal injection. During the induction period, the intravitreal injection was administered twice a week for 2 weeks. The maintenance period was once a week until the lesion became stable and the viral load of the aqueous humor turned negative. For systemic antiviral treatment, the hematologist in charge adjusted the drug regimen according to the systemic CMV infection, and regularly monitored the liver and kidney functions. For the patients with CMVR recurrence, intravitreal injections of ganciclovir were reinitiated. For the patients with poor response of fundus lesions after treatment, intravitreal injections of ganciclovir were combined with foscarnet sodium.

### 2.3. Statistical Analysis

SPSS software (version 24.0, Chicago, IL, USA) was used for statistical analysis. Descriptive data were recorded as mean ± standard deviation. *t*-tests and chi-squared tests were used to analyze the differences between groups, and Pearson’s correlation analysis was used to analyze the correlations. A *p*-value < 0.05 was considered statistically significant.

## 3. Results

### 3.1. Basic Data

During the follow-up period, the research team collected data on 52 cases (82 eyes) of CMVR after HSCT; these included 32 male (61.5%) and 20 female patients (38.5%). Of the 52 cases, 21 were monocular and 31 were binocular. The average age of patients was 27.0 ± 15.4 years (4–71 years). The best-corrected visual acuity of all patients at onset was 0.50 ± 0.39 (HM–1.0, LogMAR visual acuity: 0.71), and the best-corrected visual acuity of all patients at the treatment end point was 0.46 ± 0.36 (HM–1.0, LogMAR visual acuity: 0.69).

During the follow-up, 11 patients had recurrence during treatment, for a total of 15 eyes (recurrence rate: 21.2%). The recurrent patients included five males (45.5%) and six females (54.5%), with an average age of 30.3 ± 21.8 years (7–65 years), of which seven cases were monocular and four cases were binocular. The interval between recurrence and first onset was 6.4 ± 4.9 months (1–18 months). The initial best-corrected visual acuity of the patients with recurrence was 0.5 ± 0.4 (0.05–1.0, LogMAR visual acuity: 0.3). The best-corrected visual acuity record of the recurrent patients after treatment was 0.3 ± 0.3 (HM–1.0, LogMAR visual acuity: 0.5).

There was a significant difference between the recurrence and non-recurrence groups in the blood CD4 lymphocyte count at the onset. Other indicators, including gender ratio, unilateral and bilateral incidence ratio, onset age, initial visual acuity, final visual acuity, and viral load showed no statistical difference between groups (Table 1). The initial visual acuity of the recurrence patients was significantly correlated with recurrent visual acuity (*p* < 0.05, correlation coefficient 0.844). The patient’s visual acuity at the time of recurrence decreased significantly (*p* = 0.001). The correlation analysis of the visual acuity of the recurrence patients at onset and the lesion area at the time of recurrence had *p* = 0.064, and a correlation coefficient of −0.49 (Table 2 and Table 3).

### 3.2. CD4^+^ T Lymphocyte Count

The peripheral blood CD4^+^ T lymphocyte count of all CMVR patients at the time of onset was 133 ± 67.5/mm^3^ (40–320/mm^3^). The average CD4^+^ T lymphocyte count of the recurrent CMVR patients was 129.7 ± 80.7/mm^3^ (30–288/mm^3^). The number of CD4^+^ T lymphocytes in blood was less than 50/mm^3^ in one case of recurrence, less than 200/mm^3^ in eight cases, and more than 200/mm^3^ in two cases. The CD4^+^ T lymphocyte count had no correlation with age of onset, recurrence interval, initial visual acuity, visual acuity at recurrence, viral load in aqueous humor at recurrence, and lesion area at recurrence.

### 3.3. Detection of Viral Load in Aqueous Humor

The CMV load at the onset of all patients was 0 to 6.64 × 10^7^ copies/mL, median 7.18 × 10^3^ copies/mL. The viral load of the recurrent patients was 2.23 × 10^2^ to 3.07 × 10^7^ copies/mL, median 8.63 × 10^3^ copies/mL. Among them, four eyes had <1 × 10^3^ copies/mL, nine had >1 × 10^3^ and <1 × 10^5^ copies/mL, and the CMV DNA load of two eyes was >1 × 10^5^ copies/mL. There was no significant correlation between the viral load in aqueous humor of the recurrent patients and age of onset, recurrence interval, initial visual acuity, visual acuity at recurrence, CD4^+^ T lymphocyte count at recurrence, and lesion area at recurrence.

### 3.4. Characteristics of Fundus Lesions

After recurrence, nine eyes had fundus lesions in zone 1, eight had simultaneous optic disc and macula involvement (average visual acuity: 0.25), and one had mainly optic disc involvement. Two eyes had lesions in zone 2, one eye on the nasal side of the optic disc, and one on the temporal side of the vascular arch. One eye had lesions in zone 3. ImageJ software was used to measure the lesion and optic disc area of the recurrent patients, and the ratio of the lesion to optic disc area was recorded. The average lesion size was 62.6 ± 60.7 optic disc areas of the recurrent patients, including three eyes with >100 optic disc areas, ten eyes with >10 optic disc areas <100 optic disc areas, and two eyes with <10 optic disc areas. Recurrent fundus mainly manifested as a new yellowish fusion focus surrounding the original atrophic area, or a new small cluster of yellowish-white lesions, which may be accompanied by retinal vasculitis (Figure 1). Eight eyes had lesions involving the optic disc and macula simultaneously, one eye had retinal neovascularization, two eyes had retinal and inferior fiber proliferation, and three eyes had obvious vitreous inflammatory opacity.

The *p*-value of the correlation analysis between the lesion area and the initial visual acuity was 0.064, and the correlation coefficient was −0.49. The lesion area at recurrence was not related to the age of onset, recurrence interval, visual acuity at recurrence, CD4^+^ T lymphocyte count at recurrence, or viral load in aqueous humor at recurrence.

### 3.5. OCT Features

The OCT image of the lesion area was analyzed and described, with the scanning scope covering the original necrotic focus and the marginal recurrent focus. OCT images of recurrent lesions mainly showed diffuse hyperreflexes in the retinal neuroepithelial layer, blurred retinal layers, microcavity formation, cystoid edema within retinal layers, and exudative detachment of the retinal neuroepithelial layer, with punctate hyperreflexes in the vitreous body. During the observation and follow-up of these patients, OCT also detected liquefaction and contraction of the vitreous body due to inflammatory stimulation, as well as separation of the retinal inner limiting membrane and the vitreous posterior limiting membrane (Figure 2). Eight eyes of the enrolled patients showed punctate hyperreflexes of the vitreous body accompanied by traction of the posterior limiting membrane, and one eye showed retinal breaks on scanning. Among them, one eye underwent vitrectomy combined with silicone oil tamponade due to retinal detachment, and three eyes underwent vitrectomy to remove traction due to the aggravation of retinal proliferation.

## 4. Discussion

Cytomegalovirus retinitis is the most common opportunistic ocular infection in people with low immune function. The fundus mainly shows yellow-white granular fusion lesions with hemorrhage, accompanied by vitreous inflammation and retinal vasculitis, showing typical “cheese and tomato sauce” changes. Those most commonly affected are patients with AIDS and HSCT [10,11,12]. In recent years, many reports have been published on CMVR in patients with AIDS, but reports on CMVR after HSCT are limited, and the clinical characteristics of patients with CMVR recurrence after HSCT have not been reported.

There are many reasons for CMVR recurrence after HSCT, among which two key factors are the CD4^+^ T lymphocyte count and the CMV load in aqueous humor.

With regard to the CD4^+^ T lymphocyte count, the literature showed that the T lymphocyte subsets were significantly lower in CMVR cases within six months after HSCT (all *p* < 0.05) [13]. CMVR often occurs in patients whose CD4^+^ T lymphocyte count is less than 50/mm^3^ [6,14]. CMVR occurred more often among the recipients from alternative donors. The risk of viral infections is higher in HSCT from alternative donors including haploidentical donors and unrelated donors than human leukocyte antigen-matched sibling donors due to their severely depressed T cell-mediated immune response and insufficient resistance to the virus [15,16,17,18]. We found that the average CD4^+^ T lymphocyte count in the patients with recurrent CMVR after HSCT was significantly higher than the reference values provided in previous studies. The CD4^+^ T lymphocyte count in the recurrent group was significantly lower than that in the non-recurrent group. (1) These data show that even if the immune status of HSCT patients is not very poor, it is still possible to have a recurrence of ocular CMVR. Even if the number of CD4^+^ T lymphocytes has exceeded 200/mm^3^, stricter, more regular follow-ups should be carried out and the antiviral treatment should be supplemented in time according to the actual conditions of the patient. (2) The expert consensus on AIDS combined with CMVR treatment suggests that, if the CD4^+^ T lymphocyte count is 50–100/mm^3^, the fundus examination frequency should be once every three months and once every 6–12 months for those with a CD4^+^ T lymphocyte count greater than 100/mm^3^ [19]. This follow-up density is slightly insufficient for observing CMVR recurrence after HSCT. This study advocates that within six months after the first stabilization of CMVR in HSCT patients, the CMVR should be rechecked every 1–3 months in combination with individual immune status, and every three months from 6–12 months after the stabilization of CMVR. (3) This article summarizes the possible causes of immunosuppression in patients with CMVR recurrence after HSCT, such as immunological hemolysis and continuous multiple organ rejection. Six patients with CMVR recurrence after HSCT had a liver rejection. The existing literature also suggests that patients with liver rejection after HSCT are more likely to develop acute graft versus host reaction and have greater immune damage [19]. Acute and chronic graft versus host disease (GVHD) has been identified as an independent risk factor for the development of CMVR after HSCT for primary immunodeficiency and hematological disorders [20]. Patients with CMVR recurrence after HSCT experience more specificity in systemic and local inflammatory reactions and immune status [21]. We cannot simply follow the clinical experience of HSCT patients with CMVR at the first onset [22,23]. We should pay close attention to the changes in the disease and treat it with caution.

The CMV load in aqueous humor is also an important reason for CMVR recurrence in HSCT patients, and there is a linear relationship between the CMV load and the active lesion area of CMVR (the CMV load in aqueous humor = 3.38 + 0.01 × active lesion area) [6]. For patients with CMVR recurrence after HSCT, the CMV DNA load in aqueous humor varies greatly (10^2^ to 10^7^ copies/mL). The viral load in aqueous humor in three patients was less than 10^3^ copies/mL when they relapsed. Therefore, the definition of 10^3^ copies/mL as the standard for stopping treatment in some literature was also slightly inappropriate [16,17,24]. According to our results, even if the lesions of the CMVR patients after HSCT are completely stable and the CMV DNA in the eyes is completely negative, a considerable number of patients will still have CMVR lesion recurrence. From systemic virus infection, research indicated that even after the status of CMV antigenemia-negative had been achieved, active CMV infection remained in the retina with no symptoms, and as the CMV-infected lesions expanded, it was later recognized as CMVR [25,26,27]. Case 8 had the highest aqueous humor viral load, reaching 3.07 × 10^7^ copies/mL, the blood CD4^+^ lymphocyte count of the patient before relapse decreased from 253/mm^3^ to 195/mm^3^ within 2 months, and the patient received an intraocular dexamethasone intravitreal implant injection. These two factors suggest that while immune reconstitution is complete, intraocular hormone injection should be administered with caution under the protection of antiviral drugs, otherwise it could cause serious recurrence.

This article summarizes the characteristics of recurrent fundus lesions. The *p*-value of the correlation analysis between the initial visual acuity and lesion area at the time of recurrence was 0.064, suggesting that the wider the range of the recurrent lesions, the worse the visual prognosis was, especially when the lesions involved the optic disc and macula concurrently. In addition, CMVR recurrent lesions presented as yellow foci developing around the edge of the original focus, suggesting that the edge of the CMVR focus was the main site of inflammatory response between virus and tissue. Holland et al. also believed that the degree of opacity of the lesion margin was related to the immune status and prognosis of CMVR patients [28]. The higher the edge opacity of the lesion and the closer the lesion to the posterior pole, the greater the threat to vision, and the more active intraocular antiviral treatment was required [29].

In this study, the OCT images of the lesion area were analyzed and described. OCT can also show the changes in vitreous in the process of CMVR. The liquefaction and contraction of the vitreous body stimulated by the inflammation will stretch the necrotic lesions, forming tiny holes, which would eventually lead to retinal detachment. This study also observed one eye with tractional retinal detachment secondary to severe vitreoretinal proliferation, which differs slightly from a previous report that suggests that patients with CMVR have rhegmatogenous retinal detachment [30]. Risk factors for retinal detachment in CMVR include large areas of retinitis, bilateral involvement, and large areas of active lesions near the vitreous base [11]. Therefore, the application of OCT in clinical follow-ups is also of great significance for observing CMVR lesions. It can detect subtle changes in the vitreoretinal interface early and remind doctors to effectively intervene in high-risk patients to avoid serious complications.

By collecting and summarizing CMVR recurrence patient data after HSCT, this paper discusses the specificity of CD4^+^ T cell count, viral load, fundus photography, and OCT characteristics from different perspectives. This study had a few limitations. The number of samples collected from patients is small because of population specificity, and the medication and treatment of this systemic disease cannot be unified. This caused difficulties in case collection, creating inevitable research bias, a problem that should be considered in similar studies going forward.

## 5. Conclusions

This study suggests that the clinical features, fundus manifestations, and imaging features of CMVR recurrence after HSCT are different from those at the initial onset. Patients should be closely followed up after their condition is stable to be alert to CMVR recurrence.

## Figures and Tables

**Figure 1 jpm-13-00639-f001:**
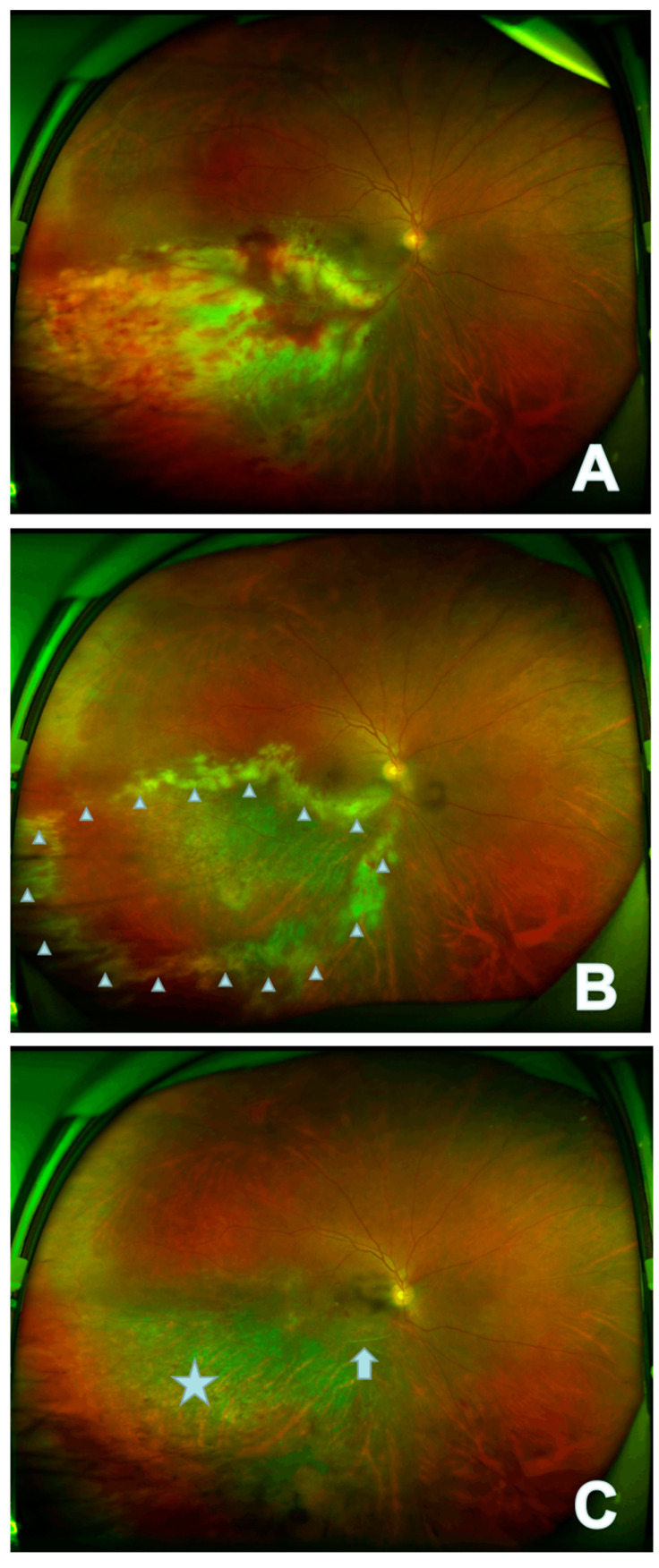
In the right eye of case 6, the lesion appears as fulminant/edematous type. The onset focus of the fundus showed a yellow-white patchy fusion lesion that had formed in the infratemporal branch of the retina with development of retinal hemorrhage (**A**); recurrent fundus mainly manifests as new yellow-white fusion around the original atrophic area (△) (**B**); the stable focus showed gray atrophy of retina (☆) with local retinal artery occlusion (↑) (**C**).

**Figure 2 jpm-13-00639-f002:**
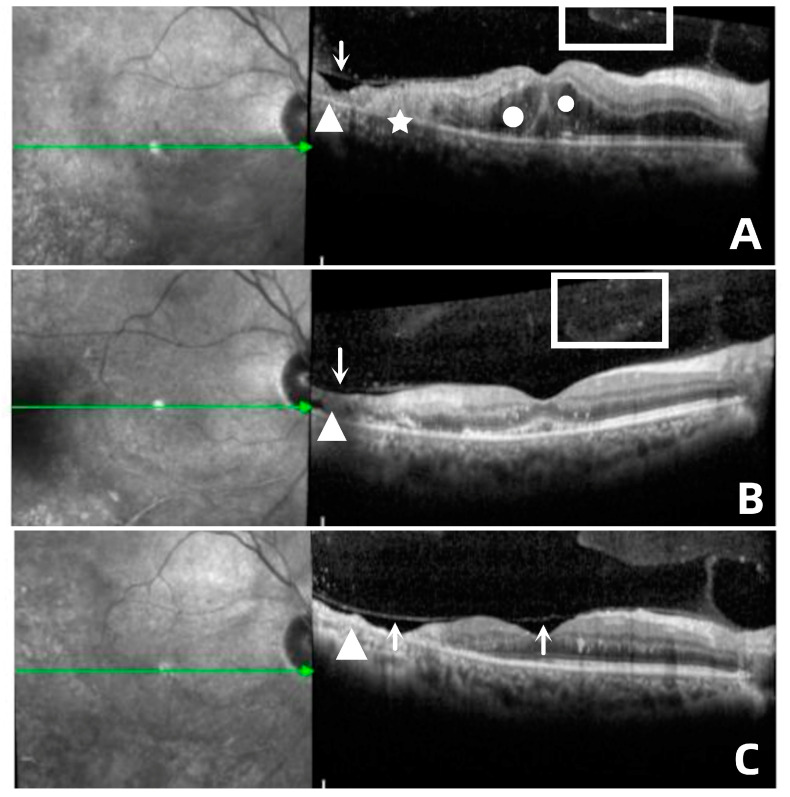
(**A**) In case 6, the recurrent focus was located at the edge of the atrophic focus (△), showing diffuse hyperreflexes (☆) across the whole layer of the retina, accompanied by multiple punctate hyperreflexes in the vitreous body (□) and macular edema (○), as well as disorder of the outer layer of the retina. (**B**) Retinal edema gradually subsided; small cavities (↓) were formed in the necrotic area, and the outer structure of retina was partially repaired after antiviral treatment. (**C**) Stable lesions showed retinal atrophy with choroidal atrophy (△), vitreous liquefaction and contraction, and separation of the inner limiting membrane and vitreous posterior limiting membrane (↑).

**Table 1 jpm-13-00639-t001:** Comparison of clinical data between recurrent and non-recurrent CMVR patients after HSCT.

Group	Number of People/Eye	Male /Female	Unilateral /Bilateral	Age (Years)	Onset Visual Acuity	Final Visual Acuity	Viral Load (Copies/mL)	Onset CD4 Count (/mm^3^)
**Non recurrent group**	41/67	27/14	14/27	26.0 ± 13.4	0.5 ± 0.4	0.5 ± 0.4	4.17 × 10^3^	295.7 ± 171.7
**Recurrent group**	11/15	5/6	7/4	30.3 ± 21.8	0.5 ± 0.4	0.3 ± 0.3	8.63 × 10^3^	129.7 ± 80.7
**Total**	52/82	32/20	21/31	27.0 ± 15.4	0.5 ± 0.4	0.5 ± 0.4	7.18 × 10^3^	243.6 ± 168.1
***p*-Value**	/	0.22	0.08	0.54	0.42	0.68	0.57	**0.00**

**Table 2 jpm-13-00639-t002:** Clinical information of patients with recurrent CMVR after HSCT.

Number	Gender	Age	Eye	Recurrence Interval (M)	Initial Visual Acuity	Recurrent Visual Acuity	CD4^+^ T Lymphocyte Count at Recurrence (/mm^3^)	Viral Load at Recurrence (Copies/mL)	Lesion Area at Recurrence (PD)	Recurrence Causes	Primary Disease
1	Male	7	Left	1	0.05	HM	191	1.08 × 10^3^	226.7	Immune hemolysis	ALL
2	Male	47	Left	5	0.1	0.05	288	1.30 × 10^2^	25.2	Persistent viremia	AML
			Right	6	0.6	0.05		3.29 × 10^4^	20.6		
3	Male	8	Right	5	0.6	0.5	30	6.42 × 10^2^	30	Liver rejection	MDS
4	Female	10	Left	14	0.3	0.1	94	1.35 × 10^4^	23.1	Liver rejection	AA
			Right	13	0.5	0.3		8.63 × 10^3^	77.3		
5	Female	10	Left	3	0.8	0.5	56	1.24 × 10^3^	65.2	Liver and intestinal rejection	MDS
6	Male	51	Right	3	0.5	0.2	72	2.08 × 10^4^	130.7	Liver and skin rejection	AML
7	Female	65	Right	3	1	0.8	105	4.36 × 10^3^	7.8	Age related immunosuppression	CML
8	Female	32	Left	9	0.3	0.1	195	3.07 × 10^7^	41.3	Continuous decrease in CD4^+^ T lymphocyte count	ALL
9	Female	58	Right	18	0.6	0.12	122	2.57 × 10^5^	18.3	Liver and Lung rejection	AML
10	Female	16	Left	3	0.5	0.3	62	2.23 × 10^2^	90.8	Skin rejection	ALL
			Right	3	0.05	0.05		8.99 × 10^2^	129.7		
11	Male	29	Left	5	1	1	212	1.39 × 10^4^	43.1	Liver rejection	AML
			Right	5	0.5	0.5		4.46 × 10^4^	8.8		

ALL Acute lymphoblastic leukemia; AML Acute myeloid leukemia; AA Aplastic anemia; MDS Myelodysplastic syndrome; CML Chronic myeloid leukemia.

**Table 3 jpm-13-00639-t003:** Statistical table of clinical data of patients with recurrent CMVR after HSCT.

Index		Age	Recurrence Interval	Initial Visual Acuity	CD4^+^ T Lymphocyte Count at Recurrence	Viral Load at Recurrence	Recurrent Visual Acuity	Lesion Area at Recurrence
Age	CC	/	0.106	0.316	0.367	0.045	0.079	−0.411
	P		0.707	0.251	0.178	0.873	0.78	0.128
Recurrence interval	CC	0.106	/	0.002	−0.024	0.152	−0.232	−0.436
	P	0.707		0.995	0.932	0.589	0.405	0.104
Initial visual acuity	CC	0.316	0.002	/	−0.15	−0.177	0.844	−0.49
	P	0.251	0.995		0.594	0.528	0	0.064
CD4^+^ T lymphocyte count at recurrence	CC	0.367	−0.024	−0.15	/	0.183	−0.148	−0.203
	P	0.178	0.932	0.594		0.514	0.598	0.469
Viral load at recurrence	CC	0.045	0.152	−0.177	0.183	/	−0.19	−0.099
	P	0.873	0.589	0.528	0.514		0.497	0.725
Recurrent visual acuity	CC	0.079	−0.232	0.844	−0.148	−0.19	/	−0.356
	P	0.78	0.405	0	0.598	0.497		0.193
Lesion area at recurrence	CC	−0.411	−0.436	−0.49	−0.203	−0.099	−0.356	/
	P	0.128	0.104	0.064	0.469	0.725	0.193	
	Mean	30.3	6.4	0.5	129.7	2073327	0.3	62.6
	SD	21.8	4.9	0.3	80.7	7919595	0.3	60.7

## Data Availability

The clinical data in this study involve patient privacy and cannot be disclosed temporarily.

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
