# Peer review of "Analysis of Clinical Characteristics of Patients with Recurrent Cytomegalovirus Retinitis after Hematopoietic Stem Cell Transplantation"

_jpm, 2023, doi:10.3390/jpm13040639_

Round 1

Reviewer 1 Report

The article by Wang et al. is a retrospective study of clinical parameters and imaging of patients with cytomegalovirus retinitis (CMVR) relapse after hematopoietic stem cell transplantation. The authors conclude by saying that several parameters of CMVR recurrence are different from those at the initial onset. Here are some of my specific comments.

1. The authors did not describe methods for CD4+ T lymphocyte count and Detection of viral load in aqueous humor.

2. The p-values for several analyses are not significant. The authors do not discuss these values or provide an explanation.

3. The article overall is well written however there are some grammatical errors that need to be corrected.

Author Response

1.The authors did not describe methods for CD4+ T lymphocyte count and Detection of viral load in aqueous humor.

Thank you for your comments. We have supplemented the relevant content. The CMV-DNA in aqueous humor was detected by PCR, and the CD4+T lymphocyte count in patients was detected by flow cytometry.

2.The p-values for several analyses are not significant. The authors do not discuss these values or provide an explanation.

Thanks for your comments, in this paper we have statements on positive and negative results for statistical p-values. However, due to the consideration of highlighting the focus of the article, we did not expand the analysis by every p-value and only performed an in-depth analysis of the key problems and those with clear significance. It is hoped that you will understand our arrangement for the article

3.The article overall is well written however there are some grammatical errors that need to be corrected.

Thanks for your comments, we have made the correction and language modifycation.

Reviewer 2 Report

The authors studied clinical and imaging characteristics of patients 12 with cytomegalovirus retinitis (CMVR) relapse after hematopoietic stem cell transplantation. Statistical analysis was performed to study different clinical parameters. I recommend to accept this case study with subject to major revisions. Following are my comments;

Please correctly format the text in tables (table 2, 3) mentioned in the manuscript

In Fig 1,I recommend authors to mention the OD/OS of fundus, and also mark the clinical phenotype in 1A.

Please mention the prevalence of CMVR in introduction in page 1

I recommend authors to discuss the effect of CMVR on ocular pressure in their study and also check the consistency of their findings with previous reported study, if any. Also check if there is any association with age of onset of disease with the clinical phenotypes.

Please discuss findings of ERG reports form affected cases, or discuss the previously reported ERG findings in CMVR cases, if reported.

Please mention the units for “leision area at occurance” heading in table 2.

Author Response

1.Please correctly format the text in tables (table 2, 3) mentioned in the manuscript

Thanks for your comments, we have made the correction

2.In Fig 1,I recommend authors to mention the OD/OS of fundus, and also mark the clinical phenotype in 1A.

Thank you for your comments. We have supplemented the relevant content.

3.Please mention the prevalence of CMVR in introduction in page 1 

Thank you for your comments. We have supplemented the relevant content at the Introduction part.

4.I recommend authors to discuss the effect of CMVR on ocular pressure in their study and also check the consistency of their findings with previous reported study, if any. Also check if there is any association with age of onset of disease with the clinical phenotypes.

Thank you for your comments. In our previous analysis, there was no significant difference in the age of onset and clinical classification. Regarding intraocular pressure, it was observed that the intraocular pressure of patients would decrease compared to the normal population, but there was no direct difference between the recurrent group and the non recurrent group.

5.Please discuss findings of ERG reports form affected cases, or discuss the previously reported ERG findings in CMVR cases, if reported.

Thank you for your opinion, while ERG is not a necessary examination for our  patient during diagnosis and treatment. And our patients are all patients after HSCT in poor physical condition. We have not introduced too many unnecessary tests. I hope you can understand.

6.Please mention the units for “leision area at occurance” heading in table 2.

Thank you for your comments. We have supplemented the relevant content.

Reviewer 3 Report

I have the following concerns, where the authors have to cover in their revision,

1- Did the author consider whether the patients are under corticosteroids, antibiotic, or other antiviral therapies during the study

2- The authors have to unify the format of all the tables, I can see that the 3 tables especially table 2 have different formats regarding font size, .....

3- Did the authors try to quantify fundus lesion using the photos, I think it would be nice if they could 

4- I recommend that the authors make use of those 2 references for quantifying eye lesions (doi: 10.3390/antibiotics11101374 and  doi: 10.3390/microorganisms9061131)

5- The manuscript requires language editing and grammar check 

Author Response

1-Did the author consider whether the patients are under corticosteroids, antibiotic, or other antiviral therapies during the study

Thanks for your comments, our patients had more complicated general conditions, and the systemic medication were all adjusted by hematologists. We as ophthalmologists did not adjust the systemic medication because of eye diseases. Hope you can understand our situation.

2-The authors have to unify the format of all the tables, I can see that the 3 tables especially table 2 have different formats regarding font size, .....

Thanks for your comments, we have made the correction

3- Did the authors try to quantify fundus lesion using the photos, I think it would be nice if they could  

4- I recommend that the authors make use of those 2 references for quantifying eye lesions (doi: 10.3390/antibiotics11101374 and  doi: 10.3390/microorganisms9061131)

We especially thank you for your comments and we can complete a new thesis if the content of lesion quantification can be realized, while we think that part is not relevant to the focus of this article. It is hoped that you will understand our arrangement for the article

5- The manuscript requires language editing and grammar check 

Thanks for your comments, we have made the correction and language modifycation.

Round 2

Reviewer 1 Report

n/a

Reviewer 3 Report

I think the authors covered my concerns, and I would accept the manuscript in its present form.